# Untargeted Metabolomics on Skin Mucus Extract of *Channa argus* against *Staphylococcus aureus*: Antimicrobial Activity and Mechanism

**DOI:** 10.3390/foods10122995

**Published:** 2021-12-04

**Authors:** Weijun Leng, Xiaoyun Wu, Tong Shi, Zhiyu Xiong, Li Yuan, Wengang Jin, Ruichang Gao

**Affiliations:** 1School of Food and Biological Engineering, Jiangsu University, Zhenjiang 212013, China; 2112118004@stmail.ujs.edu.cn (W.L.); 2221818040@stmail.ujs.edu.cn (X.W.); 1000005720@ujs.edu.cn (T.S.); xiongzhiyu@ujs.edu.cn (Z.X.); 1000003432@ujs.edu.cn (L.Y.); 2Bio-Resources Key Laboratory of Shaanxi Province, School of Biological Science and Engineering, Shaanxi University of Technology, Hanzhong 723001, China; jinwengang@nwafu.edu.cn

**Keywords:** skin mucus, *Staphylococcus aureus*, untargeted metabolic profile, antibacterial mechanism

## Abstract

Microbial contamination is one of the most common food safety issues that lead to food spoilage and foodborne illness, which readily affects the health of the masses as well as gives rise to huge economic losses. In this study, *Channa argus* was used as a source of antimicrobial agent that was then analyzed by untargeted metabolomics for its antibacterial mechanism against *Staphylococcus aureus*. The results indicated that the skin mucus extract of *C. argus* had great inhibitory action on the growth of *S. aureus*, and the morphology of *S. aureus* cells treated with the skin mucus extract exhibited severe morphological damage under scanning electron microscopy. In addition, metabolomics analysis revealed that skin mucus extract stress inhibited the primary metabolic pathways of *S. aureus* by inducing the tricarboxylic acid cycle and amino acid biosynthesis, which further affected the normal physiological functions of biofilms. In conclusion, the antimicrobial effect of the skin mucus extract is achieved by disrupting cell membrane functions to induce an intracellular metabolic imbalance. Hence, these results conduce to amass novel insights into the antimicrobial mechanism of the skin mucus extract of *C. argus* against *S. aureus*.

## 1. Introduction

The genus *Staphylococcus*, a type of classical and ubiquitous Gram-positive bacteria, are representatives of major communities that settle and live in the skin and mucous membrane of animals [1], among of which *Staphylococcus aureus* is regarded as the primary determinant for causing a variety of human clinical conditions, such as self-remissive skin infections and life-threatening syndromes [2,3]. Moreover, for a long time, clinical practice has been dealing with problems caused by this coagulase-positive bacterium in view of its potent ability and propensity to acquire resistance to antibiotics [4]. To date, it has been reported that incidences of *staphylococcal* antibiotic-resistant infections have occurred with increasing frequency in community settings [5].

Antibiotic resistance is a worldwide health care problem [6], and the application of natural antimicrobial material has gradually replaced synthetic chemical food additives in the food industry due to consumer-directed health care [7]. Therefore, the development of natural and safe alternatives to existing antibiotics against *S. aureus* infections is still of interest [8].

One source of novel antibacterial agents is fish, which have evolved to produce numerous biologically active chemicals. For instance, fish-egg lectin has been identified in several teleost species and proven to play important roles in the innate immune system against pathogen infection [3]. Bo et al. [9] first analyzed six antimicrobial peptides from a grouper fish by liver transcriptomics technology, including hepcidin 1, liver-expressed antimicrobial peptide 2 (LEAP-2), piscidin, moronecidin, NK-lysin, and β-defensin. In addition, fish permanently co-exist with various extraordinary external hazards, such as pathogens, viruses, chemical contaminants, and other segments. Therefore, the mucus, which is complex and rich in antibacterial components, such as lysozymes, immunoglobulins, and agglutinins, could act as the first defense barrier against germs. These compounds play very important roles in discriminating pathogenic microorganisms and the commensal microorganisms that can guard fish against invading pathogens [10]. Therefore, by further studying its antibacterial mechanism, skin mucus can be utilized as a source of new antibacterial substances. Microbial metabolomics is the qualitative and quantitative analysis of microbial metabolites, which currently has a crucial role in analyzing the dynamic phenotype of diverse microorganism systems and is a potent tool for revealing the antimicrobial mechanism in the field of medical microbiology. Previous studies have shown that the growth of Botrytis cinerea could be restrained by tea tree oil that can trigger mitochondrial dysfunction and oxidative stress mainly by disrupting the TCA cycle and fatty acid metabolism, and *S. aureus* treated with antibiotics generated an intricate series of metabolic alternations [11,12]. Liu et al. [13] uncovered the antibacterial mechanism of astringent persimmon tannins against methicillin-resistant *Staphylococcus aureus* isolated from pork by combining transcriptome and metabolomics analysis. 

In this paper, we prepared a skin mucus extract from *C. argus* and evaluated its inhibitory effect on *S. aureus*. Untargeted metabolomics and multivariate analysis were applied to reveal the antibacterial mechanism of the skin mucus extract of *C. argus* against *S. aureus*.

## 2. Materials and Methods

### 2.1. Reagents and Bacterial Strains

Fresh snakehead was purchased from a local supermarket (Zhenjiang city, Jiangsu, China). All specimens were transported alive to the laboratory and placed in clear water. *Staphylococcus aureus* was preserved in our laboratory and stored in 30% glycerol at −80 °C. NAD-Malic dehydrogenase (NAD-MDH), isocitrate dehydrogenase (ICDHm), α-ketoglutarate dehydrogenase (α-KGDH) reagents were purchased from Solarbio Science & Technology Co., Ltd. (Beijing, China), and Succinate dehydrogenase (SDH), adenosine 5′-triphosphate (ATP), Na^+^/K^+^-ATPase assay kits were obtained from Jiancheng Bioengineering Institute (Nanjing, China).

### 2.2. Skin Mucus Collection

Skin mucus preparation was initiated following the procedures reported by Go et al. [14]. *C. argus* was wounded by removing skin scales with sandpaper, and the mucus was collected nonlethally by anesthetizing the fish with 50 mg L^−1^ MS-222 (Sigma, St. Louis, MI, USA). Skin mucus was collected by lightly scraping the dorsal–lateral surface of the wounded *C. argus* with a plastic spatula. Then, the mucus sample was placed in a water bath and boiled for 10 min to inactivate the endogenous hydrolytic enzyme, and the boiled sample was cooled and homogenized on wet ice for 2 min. After homogenization, the sample was centrifuged (9000× *g*, 30 min, 4 °C), and the supernatant was pooled and freeze dried (−80 °C, 48 h).

### 2.3. Antimicrobial Activity

#### 2.3.1. Agar Well Diffusion Method

The antimicrobial activity of the skin mucus extract against *S. aureus* was evaluated by using the agar well diffusion method [15]. Briefly, 0.1 mL of the indicator bacteria culture (10^6^–10^7^ CFU/mL) was spread on Luria–Bertani (LB) agar medium plates. Three sterilized Oxford cups were placed on a sterile dish. After the agar solidified, the Oxford cups were removed, and 100 μL of the skin mucus extract (5, 25, 50 mg/mL) was added to the resulting hole on the dish. Sterile ultrapure water was used in place of the skin mucus extract as a negative control. Then, all the plates were put at 37 °C and cultivated for 12 h, and a digital caliper was used to determine the inhibition zone diameters surrounding the wells (including the wells). 

#### 2.3.2. Growth Curve Assay

*S. aureus* cells were collected in the mid-logarithmic phase, washed thrice, and resuspended in PBS buffer (0.01 M, pH 7.2). The resuspended solution (100 μL, 1 × 10^6^ CFU/mL) was incubated in 96-well plates with different concentrations (0, 5, 25, and 50 mg/mL) of skin mucus solution at 37 °C for 12 h with gentle shaking. 

### 2.4. Determination of Leakage of Intracellular Proteins

The protein leakage of intracellular compounds was measured to determine the antimicrobial mechanism of the skin mucus extract [16]. In brief, 1 × 10^6^ CFU/mL bacterial solution was mixed with 25 and 50 mg/mL skin mucus extract and cultured at 37 °C for 4 h. Then, supernatants containing intracellular compounds were separated by centrifugation at 6000× *g* for 5 min and measured at OD_570_ by a microplate reader (Multiskan FC, Thermo, Waltham, MA, USA).

### 2.5. Determination of Cell Membrane Damage

Bacterial cell membrane damage was assessed by measuring the release of cytoplasmic β-galactosidase from o-nitrophenyl-β-galactosidase (ONPG) (Sigma, St. Louis, MI, USA) into o-nitrophenol (o-nitrophenol) [17]. *S. aureus* was cultured in fresh LB broth containing 5% lactose, and the logarithmic phase cells were amassed in PBS (0.01 M, pH 7.2). Thereafter, the bacterial solution (60 μL) was incubated with 30 mM ONPG (10 μL) and 25 mg/mL mucus extract (30 μL). The absorbance value at 405 nm was determined every 2 h interval during incubation, which represented the hydrolysis of o-nitrophenol by β-galactosidase.

### 2.6. Scanning Electron Microscopy (SEM)

Effect of the skin mucus extract on *S. aureus* was observed under SEM [18]. A total of 1 × 10^6^ CFU/mL *S. aureus* was incubated with or without the skin mucus extract at 37 °C for 4 h. The cultivated biomass was fixed by using 5% (*v/v*) glutaraldehyde and then successively washed three times with 0.01 M potassium phosphate buffer (pH 7.2), and sequentially dehydrated with different concentration gradients of ethanol (50%, 70%, 90%, and 100%). After freeze drying at −80 °C, the samples were coated with gold and observed by SEM.

### 2.7. Laser Scanning Confocal Microscopy (CLSM)

The test strain was cultured and collected as the way described in Section 2.6. The bacterial solution (10 μL) and stain (SYTO9:PI = 1:1, 5 μL) were mixed together and reacted for 10 min on a clean glass slide for 10 min before observing by CLSM [19]. The excitation wavelength and emission wavelength of SYTO9 were 480 nm and 500 nm, respectively. The excitation wavelength and emission wavelength of PI were set as 490 nm and 635 nm, respectively.

### 2.8. Effects of Skin Mucus on ATP Production and Na^+^/K^+^-ATPase Activity

The *S. aureus* cells in the mid-logarithmic phase were collected and incubated with 25 mg/mL of the skin mucus extract at 37 °C for 4 h. Then, the cells were centrifuged at 6000× *g*, 4 °C for 10 min to harvest the supernatant. The intracellular ATP concentration and Na^+^/K^+^-ATPase activity were determined by kits according to the manufacture’s protocols.

### 2.9. Activities of Key Enzymes in the TCA Cycle

The preparation of *S. aureus* cells was conducted as described in Section 2.8. After incubation, the cells were centrifuged at 6000× *g*, 4 °C for 10 min. Then, the activities of key enzymes (ICDH, α-KGDH, SDH, and MDH) in the test cells were analyzed by spectrophotometry using commercially available kits. 

### 2.10. Sample Preparation for UHPLC–MS/MS

*S. aureus* was cultured at 37 °C for 6 h after adding the skin mucus extract. For the control, the skin mucus extracts were replaced by the same volume of sterile water. The control and treated samples (5 mL of 1 × 10^6^ CFU/mL culture) were centrifuged at 10,000× *g* and 4 °C for 10 min to acquire cell precipitates, which were subsequently washed by sterile PBS (0.01 M, pH 7.2) and collected for metabolomics analysis. One milliliter of methanol (100%) was added to the cell pellet, mixed well, and then vortexed for 1 min after sonication. Next, the sample was centrifuged at 4 °C and 12,000× *g* for 15 min, and the supernatant (10 μL) was added to liquid chromatography vials for analysis. Six independent cultures (biological replicates) were performed for all treatments.

### 2.11. Untargeted LC–MS Method for Metabolomics Detection

Metabolomics detection was determined by Hybrid Quadrupole-TOF LC/MS/MS Mass Spectrometer (Triple TOF 5600 plus, AB Sciex Instruments, Framingham, MA, USA), which was combined with Shimadzu Ultra-High-Performance Liquid Chromatography (Nexera UHPLC LC-30A, SHIMADZU, Tokyo, Japan) and equipped with a SHIMADZU InerSustain C18 (100 × 2.1 mm, 2 µm). A solution containing acetonitrile (A) and 0.1% formic acid (B) was used for gradient elution as follows: 95% to 30% B over 7 min, 30% to 0% B over 5 min, 0% B hold for 1 min, 0 to 95% B over 1 min, and finally hold for 2 min to separate all components. The autosampler and column compartment temperatures were maintained at 4 °C and 35 °C, respectively. Additionally, the injection volume was set as 10 μL, and each sample was analyzed by UPLC–MS for approximately 16 min.

### 2.12. Data Analysis

#### 2.12.1. Statistical Analysis

All the figures were drawn in Origin 2018. A significance level of *p* < 0.05 was used to determine the differences between each sample by Tukey tests of SPSS Statistics 17.0 (SPSS Inc., Chicago, IL, USA). All experiments were repeated three times, and data results were expressed as the mean plus the standard deviation (mean ± SD).

#### 2.12.2. Metabolomics Data Analysis

The original data obtained by LC–MS were converted to the ABF format by Analysis Base File Converter software and imported into MS-DIAL 4.10 for preprocessing, including the peak extraction, noise removal, deconvolution, peak alignment, and export of a three-dimensional data matrix in CSV format. Then, the extracted peak information was compared with databases including the three libraries MassBank, Respect, and GNPS for a full library search. The three-dimensional matrix information includes the following: the sample information, retention time, mass–nucleus ratio, and mass response intensity (peak area). The multivariate statistical analysis included principal component analysis (PCA), partial least squares discriminant analysis (PLS–DA), cluster analysis, and differential metabolite screening.

## 3. Results and Discussion

### 3.1. Antimicrobial Activity

The antibacterial activity of the skin mucus extract against *E. coli* was assessed by the MIC and MBC values, and the determined values were 25 mg/mL and 50 mg/mL, respectively. As shown in Figure 1A, the antimicrobial activities of the skin mucus extract against *S. aureus* increased as the concentration of the skin mucus extract increased from 25 to 50 mg/mL. The absorptions at 595 nm of the sample treated with the skin mucus extract were significantly reduced at each growth stage, compared with those in the control (Figure 1B), indicating that the growth rate of *S. aureus* was significantly inhibited. The results revealed that *S. aureus* showed different growth rates in a dose-dependent manner under different concentrations of skin mucus extract. However, the rising trend in the late growth period may be due to the adaptability of the surviving cells to the changes in the environment and pressure. Similar research showed that the growth of human pathogenic bacteria would be suppressed in a dose-dependent manner after treatment with fish mucus extracts [20]. 

### 3.2. Determination of Intracellular Protein Leakage

The release of intracellular contents, such as proteins, is an important index for measuring the integrity of the cell membrane [21]. As shown in Figure 2A, the protein leakage of *S. aureus* in the treatment groups was significantly higher than that of the group without the addition of the skin mucus extract. In particular, the protein leakage of *S. aureus* rapidly increased to 58.37 µg/mL after 4 h of treatment with 50 mg/mL skin mucus extract. Kang et al. [22] observed that the protein leakage of *P. fluorescens* after treatment with lactobionic acid (LBA) raised rapidly within 2 h and reached the peak value of 20.98 μg/mL after 8 h. The results further indicated that the skin mucus extract may exert an antibacterial mechanism by destroying the plasma membrane and causing the leakage of macromolecular substances.

### 3.3. Determination of Cell Membrane Damage

The presence of lactose and galactose in the culture medium can induce bacterial cells to produce the intramembranous enzyme β-galactosidase, which can degrade ONPG into O-nitrophenol with an absorption peak at 405 nm and can be used to determine the permeability of cell membrane [23]. As shown in Figure 2B, the OD_405_ value of samples treated with the skin mucus extract increased significantly and did not show any discernible lag, compared with that of the control group. These results suggested that the skin mucus extract caused severe damage to the cell membrane and increased the membrane permeability of *S. aureus*. A previous study indicated that the peptide produced by *Lactobacillus* FX-6 could increase the activity of β-galactosidase in the treated group [24]. 

### 3.4. Morphology of Bacteria

Figure 3A shows that the skin mucus extract altered the morphology of *S. aureus* cells and effectively eliminated the viable count of the pathogen, which conformed to the time-kill assay. The untreated *S. aureus* exhibited a slippery surface, morphological integrity, and an intact cell membrane. In contrast, the cells of the treatment group were irregular, shrunken, and exhibited some hollowness and collapse on the surface (A2, 25 mg/mL; A3, 50 mg/mL). At the same time, *S. aureus* cells experienced some effects, such as cell rupture and lysis. Therefore, it can be speculated that the skin mucus extract may disrupt the integrity of cell membranes and promote dissolution and leakage of internal cell components. Kang et al. [22] observed that MRSA cells displayed membrane morphology distortion and intracellular contents leakage after treatment with LBA. 

### 3.5. Laser Scanning Confocal Microscopy

After staining with SYTO-9/PI, the bacteria that could be seen under CLSM were stained red and green, and some were orange, which is generally believed to be caused by the overlap of dead and living bacteria [25]. As shown in Figure 3B, the control group emitted bright green fluorescence, while the treatment group showed a distinct change in the staining pattern, and the proportion of PI-positive (red dead) cells increased significantly. Hence, it can be speculated that the inhibitory effect of the skin mucus extract on the growth of the tested strains may be caused by the killing effect on the strains. Previous studies have reported that the mechanism of action of nisin involves increasing the permeability of microbial cell membranes, which results in pore generation and cell collapse [26,27]. Although SEM and CLSM analyses partly demonstrated the antibacterial properties of the skin mucus extract, further investigation should be performed to reveal the antibacterial mechanisms.

### 3.6. Determination of ATP Production and Na^+^/K^+^-ATPase Activity

ATP is one of the few high-energy compounds that tissue cells can utilize directly, and it plays a central role in cellular energy metabolism. The results (Figure 4A) showed that the ATP content dramatically dropped from 690 to 500 μmol/gprot in the treatment group. Compared with the control group, ATP levels in the treated group were downregulated by 27.5%. Cinnamaldehyde could disrupt the metabolic pathway of citric acid and inhibit bacterial ATP synthesis [28]. Similarly, Turgis et al. [29] found that mustard essential oil reduced intracellular ATP levels. The decrease in ATP content may be caused by damage to energy metabolism or disruption of ATP synthase function in cells, which results in disordered ATP synthesis. Therefore, it can be inferred that the TCA cycle may be affected based on the ATP concentration results.

Enzymes play important roles in catalyzing various physiological metabolic processes in cells. Na^+^/K^+^-ATPase is a membrane protein and can utilize ATP to transport sodium and potassium ions under an inverse concentration gradient. As depicted in Figure 4B, the Na^+^/K^+^-ATPase activity of the treatment group was significantly improved. This outcome is probably due to the increase in membrane permeability and proton conversion caused by the disruption of the cell membrane structure [30].

### 3.7. Changes in the Enzyme Activities and Metabolites Involved in the TCA Cycle

The TCA cycle is a target connecting multiple metabolic pathways and has a significant effect on the metabolic system, which is the most efficient way for the cell to obtain energy by the oxidation of sugar or other substances [31]. Products such as succinic acid and citric acid produced in the TCA cycle have important metabolic functions, such as antiobesity effects [32,33]. In this study, the citric acid content of the treatment group was increased and converted into α-ketoglutaric acid (α-KG). Moreover, as intermediates of the TCA cycle, the level of glutamate was elevated after exposure to skin mucus extract. These results showed that α-KG was mainly converted into glutamic acid, and the skin mucus extract could restrain the TCA cycle. In previous studies, external stimuli such as alcohol and acetic acid could inhibit the EMP pathway and TCA cycle of *Saccharomyces cerevisiae* [34,35,36].

Succinate dehydrogenase (SDH), isocitrate dehydrogenase (ICDH), and α-ketoglutarate dehydrogenase (α-KGDH) are three key rate-limiting enzymes in the TCA cycle pathway. As shown in Figure 4C–F, the activities of α-KGDH, SDH, and MDH in the group treated with the skin mucus extract decreased by approximately 65.23%, 36.14%, and 79.76%, respectively. However, ICDH activity showed an increase of approximately 28.96% in the treatment group. The suppression of α-KGDH and MDH resulted in a decrease in both NADH content and ATP synthesis ability of the TCA cycle, indicating that the intracellular energy supplementation was blocked and that the TCA cycle was disrupted. NADP+ can absorb hydrogen from its metabolites to form adenine dinucleotide phosphate (NADPH), which is a pivotal factor for reproducing reduced glutathione (GSH) and has a significant role in maintaining the balance of intracellular reduced GSH in cells. The increase in ICDH and decrease in NADPH content in the treatment group indicated that more NADPH was consumed for GSH synthesis, which promoted the ability of bacterial cells to cope with oxidative stress.

### 3.8. Untargeted Metabolomics Analysis

Damage to cell membranes as well as metabolic disturbance can be conducive to bacterial death [37]. Metabolic perturbations have been hypothesized to provide a protective state in bacteria by reducing the level of cytotoxic metabolic byproducts, inhibiting antibiotic uptake, and/or slowing cellular growth [38]. Therefore, untargeted metabolomics profiling of *S. aureus* under different culture conditions was conducted to determine its metabolic characteristics. PCA was used to investigate the metabolic differences between the treatment group and the control group [39]. Figure 5A shows that the PCA completely separated the samples of the treatment group and the control group, and the six biological replicates were classified into the same category, which indicated that there was a large difference between the two groups.

To further verify and identify the metabolites that were mainly responsible for discrimination between the treatment group and the control group. PLS analysis, a supervised clustering method, was performed. The major metabolic perturbations that caused this discrimination were identified from the PLS loading plot. In the PLS model, a higher VIP value means a higher contribution of the metabolite to the differentiation, and metabolites with a VIP value > 1 suggest a major contribution to discriminating groups within PLS models [40]. As shown in Appendix A, the PLS-DA model was well constructed with high R2 and Q2 values, which indicated a good fit and satisfactory predictive ability. Moreover, the clustering results were consistent with the PCA models, and PLS also resulted in a clear distinction between the metabolites in the control and treatment groups, which further suggested that the skin mucus extract might lead to systematic changes in cell metabolism pathways (Figure 5B). 

### 3.9. Enrichment and Metabolic Pathway Analysis

KEGG was used to investigate the biological categories of the identified metabolites. These metabolites were divided into different categories based on the type of compounds and their position in the metabolic pathway, such as nitrogen metabolism, arginine, and proline metabolism, alanine, aspartate and glutamate metabolism, oxidative phosphorylation, glutathione metabolism, arginine biosynthesis, fatty acid biosynthesis, butanoate metabolism, aminoacyl-tRNA biosynthesis and two-component system, which were significantly enriched for all of the identified differential metabolites (Figure 6a). Figure 6b is a constructed heat map visualizing the changes in metabolite profiles in response to the treatment of skin mucus extract, and it indicated that the skin mucus extract treatment interfered with the normal physiological metabolism of *S. aureus*. Moreover, the great mass of amino acids had elevated levels, and the contents of other metabolites in pathways were altered to varying extents in the treatment group.

ATP production occurs in two different ways in vivo, including substrate-level phosphorylation and oxidative phosphorylation, among which oxidative phosphorylation generates more high-energy phosphate bonds and is the main pathway to produce ATP. In oxidative catabolism, such as with carbohydrates and lipids, with a few exceptions, almost all ATP is generated through phosphorylation of the respiratory chain. Therefore, oxidative phosphorylation is the main source of energy required for physiological activities. In this study, after treatment with skin mucus extract, the decrease in intracellular ATP content and the elevation of ATP consumption led to a decline in the intracellular oxidative phosphorylation rate. In addition, succinate dehydrogenase and cytochrome C, as components of the respiratory chain, were found to decrease in the treatment group, which also indicated that the phosphorylation of the respiratory chain was damaged.

Normally, the glycolysis pathway, the TCA cycle, and oxidative phosphorylation are coordinated. The glycolysis pathway metabolizes the corresponding glucose into pyruvate, which is then oxidized and decarboxylated to acetyl–CoA to satisfy the requirements of the TCA cycle. The coordination between the glycolysis speed and TCA cycle rate is dependent on the inhibition of high ATP and NADH and is also regulated by citric acid. In addition, the rate of oxidative phosphorylation plays a considerable role in the operation of the TCA cycle. If the hydrogen generated by the TCA cycle cannot be effectively oxidized by phosphorylation, NADH+H+ and FADH2 will remain in the reduced state, and the dehydrogenation reaction in the TCA cycle will not proceed. Hence, interfering with the TCA cycle can also affect the glycolysis pathway and oxidative phosphorylation process to a certain extent, which is consistent with the above analysis.

Pathway analysis showed that the metabolic pathways of amino acids in the treatment group were disturbed, including arginine biosynthesis, metabolism of alanine, glutamic acid and aspartic acid, and metabolism of arginine and proline. In particular, the biosynthesis of valine, isoleucine, tryptophan, lysine, tyrosine, and arginine increased in cells exposed to the skin mucus extract, while glycine, phenylalanine, and proline were decreased (Figure 7). The biosynthesis of aspartic acid and glutamate is significantly disrupted in the alanine, aspartic acid, and glutamate metabolic pathways. When the cellular energy level is low, the oxidative decarboxylation of glutamate is accelerated, and more NADH+ and ketoglutarate acid are produced, which participate in the oxidation function. Therefore, the decrease in L-glutamate acid may affect the normal operation of the TCA cycle and the electron transport chain, resulting in the disturbance of the intracellular energy supply.

Ornithine, L-arginine and urea are all involved in the urea cycle. L-arginine is a semi-essential basic amino acid in living organisms and an important metabolite in the urea cycle that can convert high concentrations of ammonia into urea. In prokaryotic cells, the biosynthesis of arginine involves the catalysis of eight consecutive enzymes, and the initial synthesis of arginine involves acetylglutamate, which is important for most prokaryotes [41]. After acetylglutamate is synthesized, three enzymes are used to further synthesize the acrylate intermediate until the acetyl group is removed to produce ornithine, which is aminoacylated to produce citrulline, and aspartic acid is then involved in the synthesis of arginine succinate to produce arginine as the final product [42]. In the metabolic profile, it was observed that the contents of ornithine, urea, and aspartic acid were all increased significantly in the treatment group, and the content of arginine in the final product was significantly changed. This result indicated that the skin mucus extract interfered with the original balance of the synthesis of arginine and the urea cycle in *S. aureus.*

The products formed after decarboxylation have different physiological functions in organisms. For example, histidine can be decarboxylated to histamine, and ornithine and arginine can be decarboxylated to γ-aminobutyric acid, which can accelerate cell proliferation. In the treatment group, a significant decrease in histamine and the accumulation of ornithine and arginine were observed, which indicated that the skin mucus extract may disrupt the normal decarboxylation of amino acids. Furthermore, the increase in L-methionine might indicate that there is a lower requirement for L-methionine in the trans-sulfur and glutathione pathways.

By analyzing the metabolites differences, it was found that the intermediates of aminoacyl-tRNA biosynthesis, such as L-valine, L-leucine, L-isoleucine, L-methionine and L-proline, changed significantly. Amino acids must be activated to obtain additional energy before being incorporated into the peptide chain, and the activation reaction is carried out under the catalysis of aminoacyl-tRNA synthetase. The activated amino acids and tRNA form aminoacyl-tRNA, which ensures the accurate translation of the genetic code and plays an important role in protein synthesis. Aminoacyl-tRNA is an important part of the cell wall that can give it strength and shape [30]. Therefore, it was speculated that the rupture of the membrane structure may be due to the destruction of aminoacyl-tRNA syntheses, which led to the inhibition of amino acid activation and protein synthesis.

## 4. Conclusions

An untargeted LC–MS-based metabolomics approach was used to elucidate the antimicrobial metabolism mechanism of the skin mucus extract of *C. argus*. These results demonstrated that the skin mucus extract of *C. argus* had good antimicrobial activity against *S. aureus*. In addition, untargeted metabolomics analysis showed that the destruction of the tricarboxylic acid cycle and the downregulation of amino acid biosynthesis could interrupt the energy metabolism and biofilm structure, which provided novel insight into understanding the metabolic changes of *S. aureus* at the molecular level. Moreover, this study offers a theoretical basis regarding the utilization of the processing byproduct of *C. argus*, such as preservative for aquatic products.

## Figures and Tables

**Figure 1 foods-10-02995-f001:**
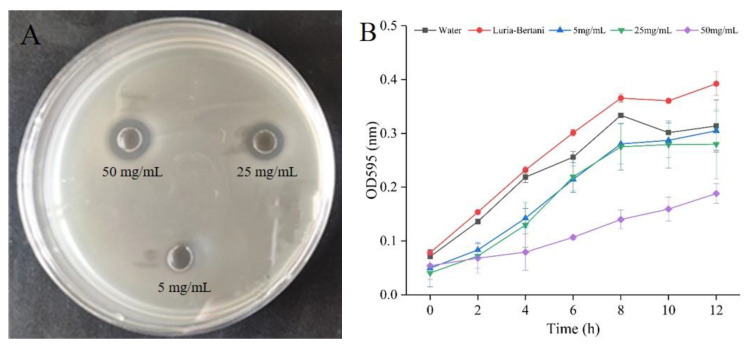
Antibacterial activity of the skin mucus extract against *S. aureus*: (**A**) zone of inhibition; (**B**) growth curves of *S. aureus* treated with different concentrations of the skin mucus extract.

**Figure 2 foods-10-02995-f002:**
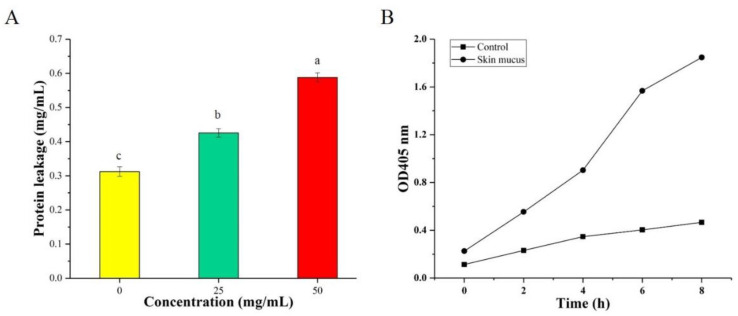
Effect of the skin mucus extract of *C. argus* on the cell membrane permeability of the tested *S. aureus*: (**A**) leakage of protein from *S. aureus* treated with the skin mucus extract; (**B**) the release of cytoplasmic β-galactosidase from *S. aureus* bacteria after treated with the skin mucus extract. Letters (a, b and c) indicated the significance between each group.

**Figure 3 foods-10-02995-f003:**
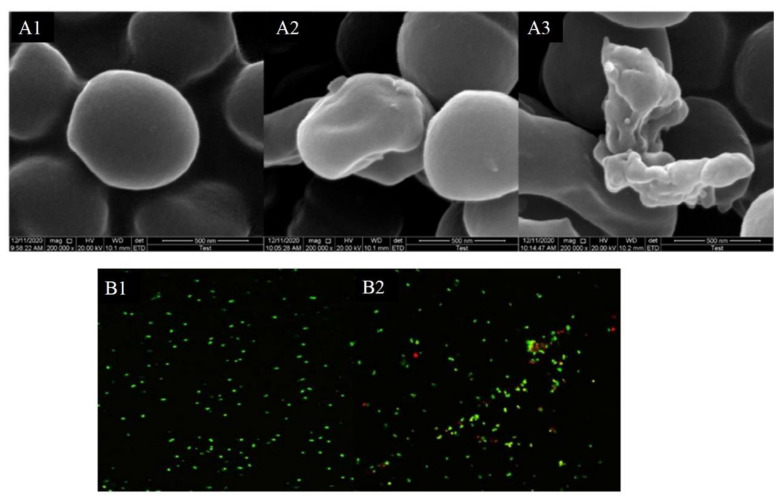
Observation of morphology changes in *S. aureus* by SEM (**A**) and CLSM (**B**). Note: (**A1**) *S. aureus* only (control group); (**A2**) *S. aureus* treated with 25 mg/mL skin mucus extract; (**A3**) *S. aureus* treated with 50 mg/mL skin mucus extract; (**B1**) *S. aureus* only (control group); (**B2**) *S. aureus* treated with 25 mg/mL skin mucus.

**Figure 4 foods-10-02995-f004:**
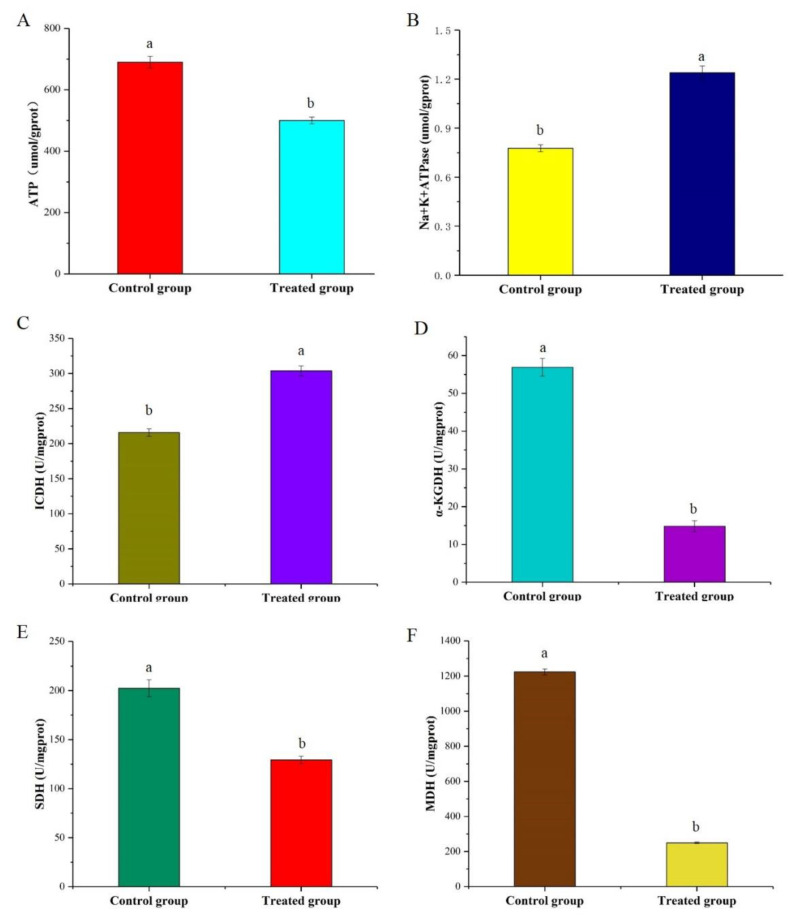
Effects of the skin mucus extract on intracellular ATP (**A**), Na^+^/K ^+^ ATPase activity (**B**), ICDH activity (**C**), α-KGDH activity (**D**), SDH activity (**E**), and MDH activity (**F**) of *S. aureus*. Letters (a and b) indicated the significance between each group.

**Figure 5 foods-10-02995-f005:**
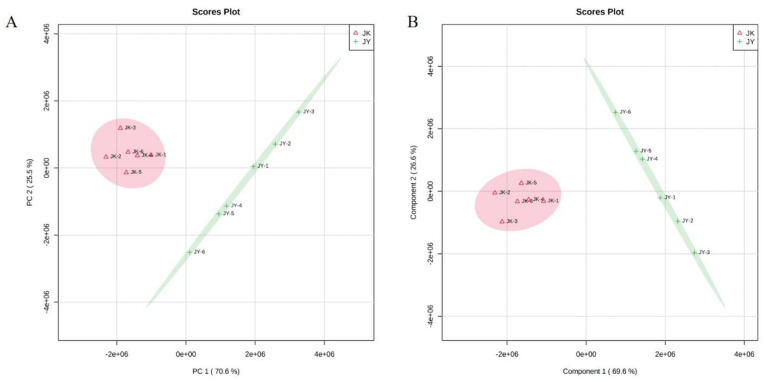
Multivariate cluster analysis of metabolite profiles of *S. aureus*: (**A**) PCA score plot; (**B**) PLS–DA score plot.

**Figure 6 foods-10-02995-f006:**
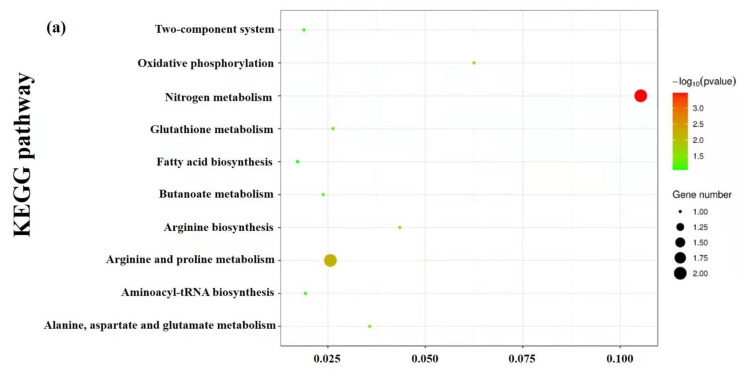
KEGG pathway analysis (**a**) and heat map (**b**) of the metabolites. The color represents the metabolite abundance (red: highest; blue: lowest).

**Figure 7 foods-10-02995-f007:**
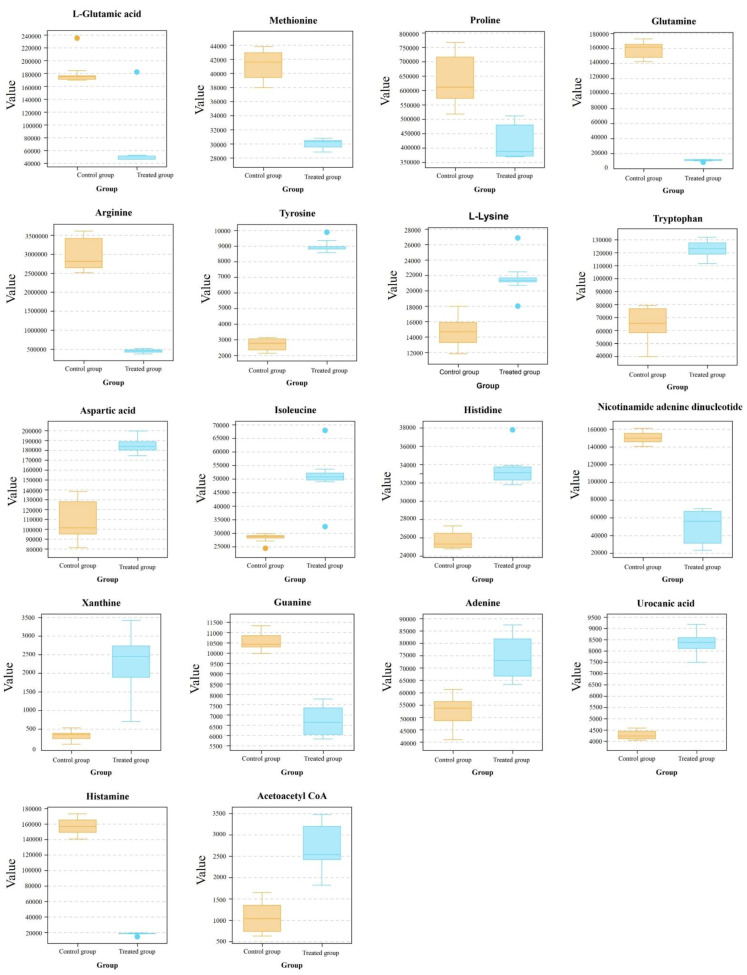
The boxplots of amino acid metabolism, purine, and pyrimidine metabolism in *S. aureus*.

## Data Availability

Data are contained within the article.

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
