# Peer review of "Untargeted Metabolomics on Skin Mucus Extract of Channa argus against Staphylococcus aureus: Antimicrobial Activity and Mechanism"

_foods, 2021, doi:10.3390/foods10122995_

Round 1

Reviewer 1 Report

The paper deals with a study on the antibacterial mechanism against Staphylococcus aureus exhibited by skin mucus extract of Channa argus. The work involved a number of biological evaluation of the antimicrobial activity of the tested substrate. These are also buttressed by scanning electron microscopy assessed direct morphological damage for the S. aureus cells subject to treatment and UHPLC-MS metabolomics studies on cellular lysates. The work could be interesting for the researchers in the specific topic of natural antibacterials. Nevertheless, since it has been submitted to a Foods related journal the connection with a possible use of the focused bio-drug in the food field (as possible food preservative ? as an example of usable antibacterial mechanism achievable with different drugs?) should be better explained and described in the manuscript. Moreover, some points of the metabolomic analysis require further clarification. The amount of used cells for the lysates preparation is missing and the chromatographic conditions should be better described (which gradient?). Please note also that the UHPLC MS described in the methods mysteriously became GC MS in the conclusions. The model parameters for the reported PCA and PLSDA are missing. The discriminating  metabolites reported in Fig 7  are differentiated by an unknown and not better specified “concentration” (Please explain if number refer to total ion current or else) Finally the heat map of the metabolites detected of Fig. 6 is almost unreadable since no description is given of metabolites and/or samples.

Author Response

Dear Reviewer

Thank you very much for the constructive comments and suggestions for improving our manuscript. We have revised the manuscript according to these comments, and retained the traces of modification. Following are responses to your comments and suggestions.

 Response to Reviewer 1 Comments

  1. Nevertheless, since it has been submitted to a Foods related journal the connection with a possible use of the focused bio-drug in the food field (as possible food preservative? as an example of usable antibacterial mechanism achievable with different drugs?) should be better explained and described in the manuscript.

   - Thank for your comments, we have added some descriptive connections with bio-drug (line 241-245, 386-389), and we have conducted a preservation experiment regarding the skin mucus extract prolonging the shelf-life of fish fillets in a follow-up experiment (Article in press).

  1. The amount of used cells for the lysates preparation is missing and the chromatographic conditions should be better described (which gradient?).

- Authors’ answer: The amount of used cells for the lysates preparation was 5 mL of 10Í6 CFU culture (line 152). And as suggested, we revised this paragraph and added more chromatographic conditions in the methodology section (line 165-166).

  1. Please note also that the UHPLC MS described in the methods mysteriously became GC MS in the conclusions.

- We are very sorry for our incorrect writing. We have corrected it (line 420).

  1. The model parameters for the reported PCA and PLSDA are missing. 

- As suggested, we have improved the resolution of Fig. 5

  1. The discriminating metabolites reported in Fig. 7 are differentiated by an unknown and not better specified “concentration” (Please explain if number refer to total ion current or else).

- Thanks for your comments, the ordinate number refer to total ion current. We made modifications to Fig. 7 and re-uploaded it.

  1. The heat map of the metabolites detected of Fig. 6 is almost unreadable since no description is given of metabolites and/or samples.

- So sorry for our negligence. We have made appropriate modifications and re-uploaded Fig. 6.

With best wishes,

Yours sincerely,

Ruichang Gao

Reviewer 2 Report

the manuscript titled "Untargeted metabolomics on skin mucus extract of Channa argus against Staphylococcus aureus: Antimicrobial activity and mechanism" by Weijun et al.,

Minor comments:

Mucous extraction: How the freeze dried extract was diluted before the anti-microbial assay

Provide the reference for the extraction method used? As well as for all the methodology used...

Author Response

Dear Reviewer

Thank you very much for the constructive comments and suggestions for improving our manuscript. We have revised the manuscript according to these comments, and retained the traces of modification. Following are responses to your comments and suggestions.

 Response to Reviewer 2 Comments

  1. Mucous extraction: How the freeze-dried extract was diluted before the anti-microbial assay.

- Authors’ answer: the freeze-dried extract was dissolved in sterile water and filtered with a 0.22 μm syringe filter, and then the high concentration solution continues to be diluted with sterile water before use.

  1. Provide the reference for the extraction method used? As well as for all the methodology used...

- Considering the Reviewer’s suggestion, all methodology used in this research had been referenced (line 83, line 117, line 124, line 133).

With best wishes,

Yours sincerely,

Ruichang Gao

Round 2

Reviewer 1 Report

Please note that the line numbering in the revised manuscript does not correspond to the line numbering on the pdf file of the revised paper. Therefore the appropriateness of the performed correction could not be checked. Moreover,  according to point 4 of the response to the reviewer 1 comments, the  model parameters for PCA and PLSDA  were asked rather than resolution of figure 5.

Author Response

Dear Reviewer

Thank you very much for the constructive comments and suggestions for improving our manuscript. We have revised the manuscript according to these comments, and retained the traces of modification. Following are responses to your comments and suggestions.

 Response to Reviewer 1 Comments (round 2)

  1. Please note that the line numbering in the revised manuscript does not correspond to the line numbering on the pdf file of the revised paper.

-Sorry for the inconvenience caused by our negligence, we rechecked the line numbering on the pdf file of the revised manuscript.

  1. The model parameters for PCA and PLSDA were asked.

-As suggested, the model parameters for PLSDA were supplied in the table below. The PLS-DA model was well constructed with high R2 and Q2 values (Table 1), which indicated a good fit and satisfactory predictive ability.

 Table 1. PLS-DA cross-validation details

Measure

1 comps

2 comps

3 comps

4 comps

5 comps

Accuracy

1.0

1.0

1.0

1.0

1.0

R2

0.90656

0.96898

0.993

0.99824

0.99941

Q2

0.8572

0.94947

0.96161

0.96433

0.96826

Response to Reviewer 1 Comments (round 1)

  1. Nevertheless, since it has been submitted to a Foods related journal the connection with a possible use of the focused bio-drug in the food field (as possible food preservative? as an example of usable antibacterial mechanism achievable with different drugs?) should be better explained and described in the manuscript.

   - Thank for your comments, we have added some descriptive connections with bio-drug (line 244-250, 291-294, and we have conducted a preservation experiment regarding the skin mucus extract prolonging the shelf-life of fish fillets in a follow-up experiment.

  1. The amount of used cells for the lysates preparation is missing and the chromatographic conditions should be better described (which gradient?).

- Authors’ answer: The amount of used cells for the lysates preparation was 5 mL of 1×106 CFU/mL culture (line 180-181). And as suggested, we revised this paragraph and added more chromatographic conditions in the methodology section (line 195-196).

  1. Please note also that the UHPLC MS described in the methods mysteriously became GC MS in the conclusions.

- We are very sorry for our incorrect writing. We have corrected it (line 459).

  1. The discriminating metabolites reported in Fig. 7 are differentiated by an unknown and not better specified “concentration” (Please explain if number refer to total ion current or else).

- Thanks for your comments, the ordinate number refer to total ion current. We made modifications to Fig. 7 and re-uploaded it.

  1. The heat map of the metabolites detected of Fig. 6 is almost unreadable since no description is given of metabolites and/or samples.

- So sorry for our negligence. We have made appropriate modifications and re-uploaded Fig. 6b.

With best wishes,

Yours sincerely,

Ruichang Gao
